# Anti-Inflammatory and Antioxidant Effects of White Grape Pomace Polyphenols on Isoproterenol-Induced Myocardial Infarction

**DOI:** 10.3390/ijms26052035

**Published:** 2025-02-26

**Authors:** Raluca Maria Pop, Paul-Mihai Boarescu, Corina Ioana Bocsan, Mădălina Luciana Gherman, Veronica Sanda Chedea, Elena-Mihaela Jianu, Ștefan Horia Roșian, Ioana Boarescu, Floricuța Ranga, Liliana Lucia Tomoiagă, Alexandra Doina Sîrbu, Andrei Ungur, Marian Taulescu, Alina Elena Pârvu, Anca Dana Buzoianu

**Affiliations:** 1Pharmacology, Toxicology and Clinical Pharmacology, Department of Morphofunctional Sciences, “Iuliu Haţieganu” University of Medicine and Pharmacy, Victor Babeș, No. 8, 400012 Cluj-Napoca, Romania; raluca.pop@umfcluj.ro (R.M.P.); bocsan.corina@umfcluj.ro (C.I.B.); abuzoianu@umfcluj.ro (A.D.B.); 2Department of Biomedical Sciences, Faculty of Medicine and Biological Sciences, “Stefan cel Mare” University of Suceava, 720229 Suceava, Romania; ioana.boarescu@usm.ro; 3Clinical Emergency County Hospital Saint John the New, 720229 Suceava, Romania; 4Experimental Centre of “Iuliu Haţieganu”, University of Medicine and Pharmacy, Louis Pasteur, No. 6, 400349 Cluj-Napoca, Romania; luciana.gherman@umfcluj.ro; 5Research Station for Viticulture and Enology Blaj (SCDVV Blaj), 515400 Blaj, Romania; chedeaveronica@yahoo.com (V.S.C.); tomoiagaliliana@yahoo.com (L.L.T.); sirbu.alexandra@ymail.com (A.D.S.); 6Histology, Department of Morphofunctional Sciences, “Iuliu Haţieganu” University of Medicine and Pharmacy, Victor Babeș, No. 8, 400012 Cluj-Napoca, Romania; jianu.mihaela21@gmail.com; 7Niculae Stăncioiu” Heart Institute Cluj-Napoca, 19–21 Calea Moților Street, 400001 Cluj-Napoca, Romania; dr.rosianu@gmail.com; 8Department of Cardiology, Heart Institute, “Iuliu Haţieganu” University of Medicine and Pharmacy Cluj-Napoca, Calea Moților Street No. 19–21, 400001 Cluj-Napoca, Romania; 9Food Science and Technology, Department of Food Science, University of Agricultural Science and Veterinary Medicine Cluj-Napoca, Calea Mănăștur, No. 3–5, 400372 Cluj-Napoca, Romania; florica.ranga@usamv-cluj.ro; 10Department of Porcine Health Management, Faculty of Veterinary Medicine, University of Agricultural Sciences and Veterinary Medicine of Cluj-Napoca, 400372 Cluj-Napoca, Romania; andrei.ungur@usamvcluj.ro; 11Department of Pathology, Faculty of Veterinary Medicine, University of Agricultural Sciences and Veterinary Medicine of Cluj-Napoca, 400372 Cluj-Napoca, Romania; marian.taulescu@usamvcluj.ro; 12Pathophysiology, Department of Morphofunctional Sciences, Faculty of Medicine, University of Medicine and Pharmacy “Iuliu Hațieganu” Cluj-Napoca, 400012 Cluj-Napoca, Romania; parvualinaelena@umfcluj.ro

**Keywords:** anti-inflammatory, antioxidant, myocardial infarction, white grape pomace

## Abstract

Grape pomace (GP), the residue left after grape pressing in winemaking, is rich in polyphenols, including flavonoids, tannins, and phenolic acids, which have antioxidant and anti-inflammatory properties. The present study aimed to evaluate the cardioprotective effects of white grape pomace (WGP) extract in two concentrations rich in polyphenols (795 mg polyphenols from WGP/kg body weight (bw) and 397.5 mg polyphenols from WGP/kg bw)), on isoproterenol (ISO)-induced myocardial infarction (MI), focusing on its anti-inflammatory and antioxidant effects. White grape pomace administration for 14 days offered a cardio-protective effect and prevented prolongation of the QT and QTc intervals on the electrocardiogram. Both concentrations of WGP prevented the elevation of nitric oxide (NO) and malondialdehyde (MDA) in the serum, with the best results being observed for the highest concentration (*p* < 0.05). White grape pomace administration offered a reduction in pro-inflammatory cytokines such as tumor necrosis factor alpha (TNF-α), interleukin 6 (IL-6), and interleukin 1β (IL-1β) in both serum and tissue in a dose-dependent manner, with the highest WGP concentration having the best effect (*p* < 0.05). Moreover, WGP reduced histological changes associated with MI. The findings of the present study demonstrate that WGP extract exerts cardio protective effects by reducing MI-associated inflammation and oxidative stress.

## 1. Introduction

Myocardial infarction (MI), commonly known as a heart attack, is a critical cardiovascular condition marked by the sudden interruption of blood flow to the heart muscle. Myocardial infarction is a leading cause of morbidity and mortality worldwide, placing a significant burden on healthcare systems and affecting millions of individuals each year [1].

Type 1 MI is generally triggered by the rupture of atherosclerotic plaques, either through rupture or erosion, and is thus classified as a result of atherosclerotic coronary artery disease (CAD); type 2 MI is characterized by ischemic myocardial injury resulting from an imbalance between oxygen supply and demand [2].

The underlying pathophysiology of MI involves a dynamic interaction of multiple factors, including inflammation, oxidative stress, and thrombosis, which contribute to the progression of atherosclerosis and the risk of plaque rupture [3]. Upon plaque rupture, a thrombus forms, obstructing blood flow and leading to the classic symptoms of MI—chest pain, shortness of breath, and, in some cases, radiating discomfort to the arm, neck, or jaw [4]. Despite advancements in medical therapies and surgical interventions, MI continues to pose a significant health risk, with high rates of recurrence and complications, especially in individuals with coexisting conditions such as diabetes, hypertension, and hyperlipidemia [5].

Isoproterenol (ISO) provides a simple, reproducible, and low-mortality method for experimental acute MI. This sympathomimetic synthetic catecholamine generates myocardial necrosis in animal models that closely resembles the lesions seen in human acute MI [6]. ISO non-selectively activates β1 and β2 receptors; given the predominance of β1 receptors in cardiac tissue, this stimulation results in positive chronotropic, dromotropic, and inotropic effects [6]. After administration, isoproterenol causes myocardial ischemia by creating an imbalance between increased oxygen demand caused by increased cardiac activity and reduced coronary blood flow [7]. ISO also alters cardiomyocyte metabolism and electrolyte balance, depletes high-energy phosphate stores, and increases oxidative stress, all of which contribute to its cardiotoxic effects and cardiomyocyte necrosis [8]. ISO is often used in animal models to study the mechanisms and potential beneficial treatments of MI [8,9,10,11,12,13,14].

Medicinal plants and phytochemicals have been extensively documented for their beneficial impacts on atherosclerosis, MI, and its associated complications [15].

Grape pomace (GP), the residue left after grape pressing in winemaking, is rich in polyphenols, including flavonoids, tannins, and phenolic acids, which have antioxidant and anti-inflammatory properties [16]. These compounds have shown potential beneficial effects in counteracting oxidative stress and reducing inflammatory markers, making grape pomace polyphenols a promising natural intervention in cardiovascular disease management [17].

However, the concentration and profile of these phenolic compounds vary significantly among grape varieties. Their synthesis and concentration depend on several factors like microclimate, genetic factors, and even vinification techniques [18,19]. Within this study, a mixture of white grape pomace (WGP) formed as a result of pressing the white grapes of different cultivars grown at Crăciunelu de Jos situated in the prestigious Târnave Vineyard, Romania, was used. Thus, the present study aimed to enhance the complexity and overall phenolic content used in this experiment and to give insights into the possible valorization of this waste, within the circular economy’s frame. This approach will therefore reflect the natural variability of local grape production and also maximize the potential health benefits and industrial applications of phenolics extracted from WGP.

Knowing that polyphenols from grape pomace may mitigate the deleterious effects of reactive oxygen species (ROS), improve endothelial function, and suppress pro-inflammatory cytokines, all of which are beneficial in the context of myocardial injury [20], this study was designed to evaluate the cardioprotective effects of WGP extract in two concentrations rich in polyphenols, on ISO-induced-MI, focusing on its anti-inflammatory and antioxidant effects.

## 2. Results

### 2.1. White Grape Pomace Characterization

The total polyphenol content (TPC) of the WGP mix extract was 194.18 ± 0.81 g gallic acid equivalents (GAE)/1 g d.w. of plant material. The high-performance liquid chromatography (HPLC) analysis identified 19 phenolic compounds (Figure 1). The extract was rich in flavanol compounds (66.5% of total phenolics), flavonols (8.4% of total phenolics), and hydroxybenzoic compounds (7.2% of total phenolics). Tannins were also present in high concentrations (17.7% of total phenolics) (Table 1). Among the compounds, catechin, epicatechin, and procyanidin dimmer IV were the major compounds represented (44.6% of total phenolics) (Table 1).

### 2.2. The Effect of WGP on ECG Parameters

The analysis of ECG parameters recorded at baseline showed no significant differences among the experimental groups; therefore, a representative ECG record for all groups is presented in Figure 2 and data are presented in Table 2.

For rats treated with saline, ISO administration induced significant alterations in ECG characteristics for MI. These changes (shown in Figure 3 and Table 3) are represented by increased heart rate (HR), prolongation of QRS complex and QT and QTc intervals, ST-segment elevation, and reduction in R wave amplitude.

Administration for 14 days of WGP and Lisinopril showed no impact on the RR interval and did not have any effect on the HR. White grape pomace prevented the prolongation of the QT and QTc intervals compared to the MI_C group. For this parameter, statistical significance was reached for MI_WGP2 group (*p* < 0.05, Table 3). White grape pomace in both doses along with Lisinopril significantly prevented R wave reduction compared to the group with MI treated with saline (Table 3).

The PR segment was not influenced by any administered substances.

### 2.3. The Effect of White Grape Pomace on Oxidative Stress Parameters

MI induction led to increased serum levels of pro-oxidant parameters such as total oxidative stress (TOS), nitric oxide (NO), malondialdehyde (MDA), and oxidative stress index (OSI) (Table 4) associated with serum levels of antioxidant parameters total antioxidant capacity (TAC) and total thiols reduction (Table 4).

Both doses of WGP prevented the elevation of NO and MDA in the serum (Table 4), with the best results being observed for the MI_WGP2 group. Results from this group were significantly lower than in group MI_LIS in terms of the above-mentioned parameters (*p* < 0.05, Table 4).

Rats from the MI_WGP2 group presented the lowest levels of TOS in the serum (Table 4), which was even lower than the negative control group and significantly lower than the group treated with Lisinopril (*p* < 0.05, Table 4). Similar results were observed for OSI.

The highest levels of total thiols were observed in the MI_WGP1 group, higher even than the negative control group; however, statistical significance was reached when it was compared to the saline-negative control group with myocardial infarction (*p* < 0.05, Table 4).

The administration of ISO led to increased serum and tissue levels for all evaluated pro-inflammatory cytokines, namely tumor necrosis factor alpha (TNF-α), interleukin 6 (IL-6), and interleukin 1β (IL-1β). Tumor necrosis factor-alpha and IL-6 were significantly increased in the MI_C group, not only in the heart tissue (*p* = 0.011 for both parameters) but also in the serum (*p* = 0.001 and 0.045 for TNF-α and IL-6, respectively) when compared to the negative C group (Figure 4 and Figure 5). Interleukin 1β was also increased in both serum and heart tissue when compared with the MI_C group, but statistical significance was not reached (*p* > 0.05, Figure 6).

The administration of white grape pomace offered a reduction in TNF-α in both serum and tissue. Rats from the MI_WGP1 group had statistically lower serum levels of TNF-α compared to the saline-negative control group (*p* < 0.044, Figure 4). Lisinopril also provided a reduction in TNF-α, especially in the serum, but this was not statistically significant (*p* > 0.05, Figure 4).

Serum levels of IL-6 were slightly influenced by the administration of WGP extract. Serum levels of IL-6 were significantly reduced after Lisinopril administration compared to the MI_WGP1 and MI_WGP2 groups (*p* = 0.005 and *p* = 0.006, respectively, Figure 5). Between MI_WGP1 and MI_WGP2 groups, the lowest tissue levels of IL-6 were observed in the MI_WGP1 group, but statistical significance was not reached (*p* > 0.05, Figure 5).

Serum and tissue levels of IL-1β were reduced after WGP administration. Both concentrations offered similar results to Lisinopril within the heart tissue (Figure 6b). As for the serum levels, the lowest level was found in the MI_WGP1 group, which was significantly lower than the group treated with Lisinopril (*p* = 0.046), the control group (*p* = 0.040), and the MI_C group (*p* = 0.007), (Figure 6a).

### 2.4. Histological Findings

Histopathological evaluation revealed distinct differences in myocardial changes among the experimental groups, as assessed using the Metias and Jokinen scoring systems.

In the control group (C), no histopathological changes were observed, with all samples scoring 0 on the Jokinen scale and showing no alterations in the Metias scoring framework (Figure 7). According to the Metias scoring system evaluation, the control was different and statistically significant when compared to all other groups (*p* ˂ 0.007), except for the MI-WGP1 group (*p* = 0.110). According to Jokinen scoring systems, the control group was statistically different only when compared to MI_C and MI_LIS (*p* ˂ 0.001).

In the myocardial-infarction-positive control group (Group 2, MI_C), significant degenerative and inflammatory changes were detected. Samples predominantly exhibited moderate (++ on Metias) to marked (+++ on Metias) myofibrillary degeneration and inflammatory infiltrates. Correspondingly, these sections scored 2–3 on the Jokinen scale. In some samples, necrosis accompanied by a diffuse inflammatory process was obvious (Figure 7).

The groups pre-treated with white grape pomace extract (MI_WGP1 and MI_WGP2) demonstrated reduced severity of histopathological changes. In Group 3 (MI_WGP1), myocardial sections primarily showed mild (+ on Metias) focal myocyte degeneration and limited inflammatory responses (*p* ˂ 0.017, as compared to MI_C), with a Jokinen score of 1 (minimal) (*p* ˂ 0.011 as compared to MI_C)) (Figure 8). Group 4 (MI_WGP2) displayed more severe changes compared to group 3 (MI_WGP1), with most samples exhibiting moderate (++ on Metias) changes and Jokinen scores between 1 and 2 (Figure 8).

The group treated with Lisinopril (Group MI_LIS) showed significant degenerative and inflammatory changes. Samples predominantly exhibited moderate (++ on Metias) to marked (+++ on Metias) myofibrillary degeneration and inflammatory infiltrates, with no statistical significance when compared to the MI_C group. Correspondingly, these sections scored 2–3 on the Jokinen scale. In some samples, necrosis accompanied by a diffuse inflammatory process was obvious (Figure 9).

These findings prove a potential protective effect of white grape pomace extract in mitigating myocardial damage, as evidenced by reduced histopathological severity compared to the MI-C group, whereas the protective effect of Lisinopril could not be demonstrated.

## 3. Discussion

It is known that both white and red GP are important sources of bioactive compounds, especially phenolic compounds like phenolic acids, flavonols, proanthocyanidins, flavonols, stilbenes, anthocyanins, and tannins [26,27,28,29]. The main differences between the two types of grape pomaces are the missing anthocyanins in WGP [30], and the different profiles and concentrations of phenolics as influenced by the environmental factors, soil, climate conditions, grapes varieties, pomace production, or fermentation conditions [31,32,33,34].

Most studies evaluated the effects of red grape pomace [35], grape seeds [36], red wine [36], or resveratrol [37,38]. Few published studies investigated the cardioprotective effects of WGP, especially in myocardial ischemia.

The present study showed that WGP pre-treatment offered cardioprotective effects in isoproterenol-induced MI. Specifically, it helped in controlling the evaluated oxidative stress parameters and pro-inflammatory cytokines. It also helped in reducing the electrocardiographic and histological changes associated with MI.

These results can be attributed to the rich composition of WGP in phenolic compounds, especially in catechin, epicatechin, and procyanidin dimmer that were present in approximately 45% of total identified phenolics. Similar results, with flavanols as the most abundant phenolic class of compounds identified in different WGP varieties cultivated in northern Spain, were also reported [39].

It is well known that flavanols interact with cell membranes by modulating fluidity and permeability, primarily through their hydroxyl-substituted aromatic rings [40]. Their protective effects against oxidative stress involve inhibiting lipid peroxidation, scavenging free radicals, and chelating redox-active metals, thereby preserving membrane integrity and function [41,42,43]. Also, these compounds exert anti-inflammatory effects by modulating key signalling pathways involved in the inflammatory response, such as nuclear factor kappa-light-chain-enhancer of activated B cells (NF-κB) and mitogen-activated protein kinase (MAPK), leading to the downregulation of pro-inflammatory cytokines and mediators [44].

The results of this study show that after ISO administration, the RR interval was reduced—and therefore HR increased—as a result of β1 and β2 adrenoreceptor stimulation by ISO, which exerts positive inotropic effects [45]. Electrocardiographic markers indicative of ISO-induced myocardial damage included prolonged QRS complex duration, lengthened QT and QTc intervals, ST-segment elevation, and reduced R-wave amplitude [46]; all of these were observed in the control group treated with saline.

The prolonged QRS complex and long QT/QTc intervals observed following ISO administration are likely due to slowed ventricular conduction caused by ISO’s cardiotoxic effects [47]. Prolongation of the QT interval is a recognized marker of increased risk for ventricular arrhythmias and sudden cardiac death [48]. The elevated heart rate induced by ISO can impair myocardial perfusion [49]. The reduced R-wave amplitude may result from myocardial edema caused by ISO, whereas ST-segment elevation reflects the loss of cardiomyocyte membrane potential and the action potential disparity between ischemic and non-ischemic myocardial regions [50]. Saline solution, often used as a placebo, is physiologically inert and does not directly influence their parameters, leaving the ISO-induced ECG changes unaffected.

In this study, WGP administration for 14 days demonstrated greater effectiveness in preventing QT prolongation compared to LIS treatment. Similar results were reported in previous research on captopril, another angiotensin-converting enzyme inhibitor (ACEI), which also prevented QT interval prolongation in ageing rats [51]. Angiotensin-converting enzyme inhibitors are known to reduce systemic vascular resistance with minimal effects on heart rate [52], explaining the lack of significant heart rate differences between MI_LIS and MI_C groups. Preservation of cardiomyocyte structure and function in rats pretreated with WGP could explain the reduction in the QRS duration after ISO administration. These findings highlight the cardioprotective potential of WGP in ISO-induced myocardial injury.

Prolonged inflammation and oxidative stress play critical roles in the onset, progression, and pathogenesis of MI [53]. Cardiac ischemia an injury induced by ISO administration in rats triggers both inflammation and oxidative stress, resulting in the production of pro-inflammatory cytokines and ROS, which cause structural cellular damage [54].

Total antioxidant capacity (TAC) is often reduced in MI patients, making antioxidant therapy a potential strategy for coronary artery disease prevention [55]. Conversely, TOS levels are elevated in patients with chronic ischemic heart failure [56]. Thiols, which serve as crucial antioxidants, play a protective role against lipid peroxidation caused by ROS [57]. A reduction in total thiols, as seen in MI patients, indicates their consumption in response to increased ROS production due to ischemia and reperfusion [58]. High OSI indexes were significantly correlated with disease severity in patients with MI [59].

Red grape pomace extract was observed to improve the antioxidant defences of TAC [60]. This effect is primarily due to its flavonoids, resveratrol, and tannins, which neutralize free radicals and inhibit oxidative damage, thus potentially reducing myocardial injury during ischemic events [20]. In the present study, WGP protected against oxidative stress by stimulating thiol production when administered in higher concentrations. Total thiols were reported to be elevated in other grape pomace extracts [61], similarly to the results from this study.

Lipid peroxidation in heart tissue was assessed through MDA, a key marker of oxidative stress [62]. Malondialdehyde, a stable byproduct of polyunsaturated fatty acid peroxidation and arachidonic acid metabolism, is another important oxidative stress marker [63]. Acute ischemic injury during MI leads to increased MDA levels due to oxidative stress and low oxygen availability [64]. Plasmatic MDA levels rise immediately after MI as a consequence of acute ischemic oxidative damage [58]. Similarly to the results from the present study, Balea et al. reported that red wine grape pomace reduced serum levels of MDA in ISO-induced myocardial ischemia [14]. Attenuation of MDA levels reduces lipid peroxidation and oxidative stress, helping to preserve myocardial tissue integrity and improving cardiac outcomes [65].

The high concentrations of NO were found in rats treated with ISO [66] which demonstrates the increase in NO synthesis as a response to myocardial infarction, with the activation of the high-output inducible NOS (iNOS)-NO pathway [67]. Elevated iNOS-derived NO contributes to peroxynitrite formation, leading to significant oxidative stress [68], myocardial apoptosis [69], and further enlargement of the infarcted area [70]. Attenuation of NO levels during MI can reduce oxidative damage by limiting harmful peroxynitrite formation, which damages cellular structures. Therapies aimed at modulating NO production, such as selective iNOS inhibitors or antioxidants, can mitigate nitroxidative stress, preserving myocardial function and promoting repair. Balancing NO levels is critical for therapeutic strategies in MI [65]. The administration of grape pomace extracts has been previously associated with reduced levels of malondialdehyde, indicating a decrease in oxidative stress [71]; in this paper, this effect was observed in MI.

Tumor necrosis factor–alpha IL-6 are key pro-inflammatory cytokines that play a critical role in collagen synthesis and scar formation following acute MI [72]. Under normal conditions, TNF-α is not expressed in cardiomyocytes; however, during acute MI, ischemia and hypoxia activate cardiomyocytes and myocardial macrophages, leading to substantial TNF-α production in the infarcted and peri-infarct zones [73]. Elevated serum levels of IL-6 after acute MI have been well-documented, with a strong correlation between high IL-6 and C-reactive protein (CRP) levels and peak cardiac troponin levels. This suggests a clear association between inflammation and the extent of myocardial injury [74]. Similarly, IL-1β levels are elevated during myocardial ischemia, contributing to the activation of myofibroblasts, which are implicated in cardiac remodelling and the impairment of systolic function post acute MI [75,76,77]. Studies have shown that reducing IL-1β levels is associated with a smaller infarcted myocardial area, highlighting its pivotal role in acute MI pathophysiology [77,78].

In the present study, WGP administration was observed to reduce TNF-α and IL-1β levels and had limited effects on IL-6.

The underlying mechanisms by which grape pomace influences TNF-α levels involve the modulation of key inflammatory pathways. Farina et al. demonstrated that extracts from grape pomace extracts inhibited TNF-α mediated NF-κB activation, thereby reducing the production of pro-inflammatory cytokines [79], while Nishiumi et al. found that a diet supplemented with red grape pomace could suppress the expression of iNOS and cyclooxygenase-2 (COX-2) through the inhibition of NF-κB activation [80]. The inhibition of IL-1β by suppressing the NF-κB by grape pomace extract was observed in several studies [81,82,83].

In an in vivo study performed by Pop et al., which evaluated the anti-inflammatory effects of white and red pomace extracts, it was observed that even though a decreasing trend in the level of IL-6 was obtained for both types, no statistical difference was reached [84]. These results are consistent with the findings of this study.

These findings suggest a potential protective effect of WGP extract in mitigating myocardial damage, as evidenced by reduced histopathological severity compared to the MI-C group. Nour et al. concluded that grape seed extract administered throughout 14 days after ISO administration offered a better histological picture with a significant decrease in the percentage of the area affected by fibrosis [85].

### Limitations and Future Directions

Some limitations can be identified in this study. For instance, echocardiographic assessments of heart function, such as global or focal hypokinesia, left ventricular end-diastolic and systolic dimensions, and ejection fraction, were not conducted. Additionally, molecular evaluations of proteins involved in pathways related to myocardial inflammation, fibrosis, and necrosis were not performed, though they would have provided valuable insights.

Future research will aim to address additional oxidative stress parameters such as superoxide dismutase (SOD), catalase, glutathione (GSH), glutathione S-transferase (GST), and glutathione peroxidase-1 (GPx-1), and role of nuclear factor erythroid-2-related factor 2 (Nrf2). The evaluation of nuclear factor kappa-light-chain-enhancer of activated B cells (NF-κB) signalling would also be of great interest for better understanding the anti-inflammatory effects of WGP. Future research should fill other gaps by incorporating echocardiographic evaluation. Furthermore, fibrosis development and the extent to which it follows MI could be better highlighted through histological techniques, such as Masson’s Trichrome staining. Further research is necessary to identify which specific ingredient or ingredients may be contributing most significantly to these cardioprotective effects.

All these evaluations will enhance the understanding of WGP’s cardioprotective effects and potential therapeutic applications.

## 4. Materials and Methods

### 4.1. Ethics Statement

All experiments were performed following the Declaration of Helsinki on Animal Studies. The experimental protocol was approved by the Ethics Committee of the “Iuliu Hațieganu” University of Medicine and Pharmacy in Cluj-Napoca and by the Veterinary Sanitary and Food Safety Directorate in Cluj-Napoca (certificate no.255/13.05.2021). The experimental protocol was in adherence with national and international guidelines for animal care and use.

### 4.2. Chemicals and Reagents

Isoproterenol hydrochloride (ISO) (98%) was purchased from Sigma–Aldrich (St. Louis, MO, USA). Ethanol, acid acetic, sodium carbonate, Folin–Ciocalteu reagent, phosphate buffer, were bought from and Merck Co. (Darmstadt, Germany) and Sigma Co. (St. Louis, MO, USA). Acetonitrile of HPLC grade and ultrapure water were provided from Merck (Germany). Gallic acid, catechin, and rutin (of 99% HPLC grade), were bought from Sigma (USA).

### 4.3. Plant Material

White grape pomaces consisted of skins, seeds, and stems, of white wine grapes (*Vitis vinifera* L.) provided by SCDVV Blaj winery (Blaj, Târnave Wine Center, România). Traminer roz, Neuburger, Sauvignon blanc, Fetească regală, Riesling Italian, Muscat ottonel and Iohaniter white grape cultivars were harvested from Crăciunelul de Jos vineyard between 12 September 2019 and 18 September 2019. The WGP was obtained by pressing the grapes, and it was dried at room temperature. After drying, the WGP was frozen at −80 °C until the extraction.

### 4.4. Grape Pomace Extraction

A mixture of equal amounts of each WGP cultivar (30 g) was ground to a fine powder and then extracted in 40% ethanol (1000 mL). The mixture was sonicated for 30 min at room temperature, followed by filtration (Whatman filter paper no. 3). After 30 min, the pellets were re-extracted and the supernatants were combined. The procedure was repeated for a total number of three times. The ethanolic extracts were concentrated to a final volume of 350 mL using a rotary evaporator (Heidolph Hei-VAP Platinum 3, Heidolph Scientific Products GmbH, Schwabach, Germany) and further analyzed for their total polyphenol content and phenolics profile.

### 4.5. Total Polyphenol Content (TPC)

The Folin–Ciocalteu method was employed to determine the total phenolic content (TPC) as outlined by Pop et al. [86]. In brief, 25 mL of WGP extract was combined with 125 mL of Folin–Ciocalteu reagent (0.2 N) and 100 mL of sodium carbonate (Na_2_CO_3_) solution (7.5% *w*/*v*). The resulting mixture was homogenized and incubated for 2 h at room temperature under dark conditions [87]. Absorbance was measured at 760 nm using a Synergy HT Multi-Detection Microplate Reader (BioTek Instruments, Inc., Winooski, VT, USA). The TPC was calculated using a gallic acid calibration curve (R^2^ = 0.9945) and expressed as gallic acid equivalents (GAE). Each sample was analyzed in triplicate, with results reported as mean values (mg/100 g dry weight (DW) of extract) ± standard deviations.

### 4.6. Phenolic Compounds Analysis by Liquid Chromatography-Diode Array Detection–Electro-Spray Ionization Mass Spectrometry (HPLC-DAD-ESI MS)

High-performance liquid chromatography (HPLC) is a validated method used to separate, quantify and identify individual compounds within a mixture [88,89]. The HPLC-MS analysis of WGP ethanolic extract was conducted as described by Pop et al. [86]. The analysis utilized an Agilent 1200 HPLC system equipped with DAD detection and coupled to a single quadrupole mass spectrometer (Agilent 6110) (Agilent Technologies, Santa Clara, CA, USA). Separation was performed at room temperature using an Eclipse XDB C18 column (4.6 × 150 mm, 5 µm particle size) (Agilent Technologies, Santa Clara, CA, USA) and a gradient of two mobile phases: (A) 0.1% acetic acid/acetonitrile (99:1) in distilled water (*v*/*v*), and (B) 0.1% acetic acid in acetonitrile (*v*/*v*) [86]. The gradient program for compound separation was as follows: 95% A from 0 to 2 min; a linear decrease from 95% to 60% A from 2 to 18 min; a further decrease to 10% A from 18 to 20 min; and finally, a return to 95% A within 1 min, maintained for an additional 5 min [87]. The flow rate was set at 0.5 mL/min. Absorbance was recorded at 280 nm and 340 nm. The separated compounds were subsequently injected into the mass spectrometer equipped with an ESI source operating in positive ion mode. The parameters included a source temperature of 350 °C, nitrogen flow of 8 L/min, and capillary voltage of 3000 V. Scanning was performed over a mass-to-charge ratio (*m/z*) range of 100 to 1000. Data analysis was conducted using Agilent ChemStation Software (Rev B.04.02 SP1, Palo Alto, CA, USA). Compound identification was achieved by integrating UV-visible spectra, retention times, mass spectra, and reference data from the literature. For quantification, three calibration curves were performed. Accordingly, standards of gallic acid, catechin, and rutin were injected five times to obtain the calibration curve. Following, compounds belonging to hydroxybenzoic acids were quantified as gallic acid equivalent (R^2^ = 0.9978; y = 33.624x + 30.8; LOD = 0.35 ug/mL, LOQ = 1.05 ug/mL); compounds belonging to flavanols were quantified as catechin equivalent (R^2^ = 0.9985, y = 15.224x − 130.24, LOD = 0.18 μg/mL, LOQ = 0.55 μg/mL ), and compounds from flavonols class as rutin equivalent (R^2^ = 0.9981, y = 26.935x − 33.784, LOD = 0.21 μg/mL, LOQ = 0.64 μg/mL)

### 4.7. Animal Grouping and Myocardial Ischemia Induction

A total of fifty male white Wistar–Bratislava rats (200–250 g) were used in this study, sourced from the Animal Department of the Faculty of Medicine at Iuliu Hațieganu University of Medicine and Pharmacy from Cluj-Napoca. Throughout the experiment, the rats were housed in polypropylene cages and allowed to acclimate to standard environmental conditions, which included a temperature range of 22–24 °C, humidity of 55 ± 15%, and a 12 h light/dark cycle. The animals had unrestricted access to standard food pellets and water was provided ad libitum.

The rats were randomly divided into five groups of ten animals each and treated as follows:(1)Group 1 (C), the control group, in which rats received only saline and no MI was induced.(2)Group 2 (MI-C), the myocardial infarction model group, in which rats were pre-treated with saline and MI was induced.(3)Group 3 (MI_WGP1), rats were pre-treated with 795 mg polyphenols from WGP/kg bw and MI was induced.(4)Group 4 (MI_WGP2), rats were pre-treated with 397.5 mg polyphenols from WGP/kg bw and MI was induced.(5)Group 5 (MI_LIS), rats were pre-treated with Lisinopril at a dose of 10 mg/kg bw and MI was induced.

A flowchart demonstrating the study groups and interventions in Figure 10.

To induce MI, rats from groups 3 to 5 were intraperitoneally injected with a single dose of ISO in a concentration of 45 mg/kg on day 13 of the experiment [9]. It was previously demonstrated that a dose of 45 mg/kg of ISO can be used to induce infarct-like lesion with biological, electrocardiographic (ECG), and histological changes characteristic of MI [10,11]. Group 1 received subcutaneously injected saline following the schedule of the groups with MI.

White grape pomace extracts were administrated orally daily, by gavage.

The persons involved in manipulating the rats, treatment administration and those who performed the biochemical and histological analysis were not aware of the study groups.

### 4.8. Electrocardiography Monitoring

Electrocardiography recordings were taken before ISO administration (on day 0) and after ISO administration (on day 14) using the method described by Boarescu et al. [8]. Animals were anaesthetized with xylazine (2.6 mg/kg, i.p.) and ketamine (26 mg/kg, i.p.), and 15 min after administration, the animals were placed in the supine position. Electrodes were attached to the paw pads of each rat, and ECG recording was performed in lead II using a Biopac MP36 system (Goleta, CA, USA) calibrated at 1 mV/1 cm and a speed of 50 mm/s. The Biopac Student Lab 3.7.7 software was used to calculate the RR intervals (ms), PR segments (ms), QRS duration (ms), QT intervals (ms), and R wave amplitude (mV). Based on the recorded data, the heart rate (HR) (beats/min) was calculated from the RR interval using this formula: HR = 60,000/RR. The modified Bazett formula: QTc = QT/√(RR/150) was used to calculate the corrected QT interval (QTc) (ms) [90].

### 4.9. Blood Samples

On day 14, 24 h after the dose of ISO was administered, blood samples were taken from the retro-orbital plexus of each rat, under mild sedation (ketamine 20 mg/kg b.w., i.p. and xylazine 2 mg/kg b.w., i.p.) [91]. At the end of the experiment, all rats were euthanized using an overdose of anaesthetic agent.

### 4.10. Tissue Homogenate

Heart tissue samples were harvested in all groups immediately after sacrification. The samples were weighed and homogenized in four volumes of phosphate-buffered saline using an automated Witeg Homogenizer (HG-15D, Wertheim, Germany). The homogenate was then centrifuged at 1500× *g* for 15 min at 4 °C, and the resulting clear supernatant was stored for subsequent biochemical analyses.

### 4.11. Blood Samples and Serum Analysis

Pro-inflammatory cytokines, including tumor necrosis factor-alpha (TNF-α) (Catalog no 900-K73), interleukin 1 beta (IL-1β) (Catalog no 900-K91), and interleukin 6 (IL-6) (Catalog no 900-K86), were quantified using the ELISA method (Stat Fax 303 Plus Microstrip Reader, Minneapolis, USA) following the protocols provided by commercial ABTS ELISA development kits (PeproTech EC, Ltd., London, UK).

To evaluate oxidative stress, plasma levels of total oxidative stress (TOS), nitric oxide (NO), malondialdehyde (MDA), total antioxidant capacity (TAC), and total thiols (THIOL) were measured using a Jasco V-350 UV-VIS spectrophotometer (Jasco International Co., Ltd., Tokyo, Japan). TOS was determined using an automated colorimetric method described by Erel O [92]. The method described by Miranda et al. was used to detect the plasma NO levels [93]. MDA, a marker for lipid peroxidation, was detected using thiobarbituric acid [94]. TAC was evaluated using a colorimetric and fully automated method described by Erel O [95] and total thiols were measured after the method described by Hu M [96]. The oxidative stress index (OSI) was calculated using the formula OSI = TOS/TAC [97].

### 4.12. Histopathological Assessment

Adhering to ethical standards and pre-approved protocols, the animals were humanely euthanized. The hearts were excised, and any excess tissues, fat, and surrounding blood vessels were meticulously removed. The specimens were subsequently immersed in 10% phosphate-buffered formalin (pH 7.0) for a 24 h fixation period. After fixation, the samples were processed for standard paraffin embedding. Sections of 3 μm (µm) thickness were prepared using a microtome. These sections were stained with hematoxylin and eosin (H&E) to facilitate microscopic examination. Digital images of the stained sections were captured using a Zeiss Axiocam 208 color digital camera integrated with Zen Core v 3.0 imaging software, both manufactured by Carl Zeiss AG, Oberkochen, Germany.

The histopathological examination of heart tissues was focused on detecting degenerative and inflammatory lesions within the myocardium. Observations were categorized and evaluated using semi-quantitative scoring systems, specifically those developed by Metias and Jokinen [98,99]. In the Metias scoring framework, myocardial changes were classified into the following categories: no alterations, +mild (indicating focal myocyte damage or limited multifocal degeneration with minimal inflammatory response), ++moderate (characterized by widespread myofibrillary degeneration and/or diffuse inflammation), and +++marked (indicating necrosis accompanied by a pervasive inflammatory process). Based on the observed histopathological changes, the animals were grouped as follows: Group 1 (no histopathological changes), Group 2 (mild histopathological changes), and Group 3 (moderate to severe histopathological changes). The Jokinen system was employed to assign a numerical grade to the severity of microscopic changes, using a scale from 0 to 4: 0 (no changes), 1 (minimal), 2 (mild), 3 (moderate), and 4 (severe).

### 4.13. Statistical Analysis

Data analysis was performed using IBM SPSS Statistics software version 29.0.0.0 (IBM Corp., Armonk, NY, USA). The Shapiro–Wilk test was used to assess data distribution. Since the data showed a non-normal distribution, continuous variables were presented as medians and interquartile ranges (25th–75th percentiles). Group comparisons were made using the non-parametric Kruskal–Wallis test, with a statistical significance of *p* < 0.05.

## 5. Conclusions

Taken together, the findings of our study demonstrate that white grape pomace extract exerts antioxidant effects in myocardial infarction by suppressing oxidative stress and enhancing anti-oxidative activity. Beneficial effects can also be associated with its ability to reduce serum and tissue levels of certain pro-inflammatory cytokines. Other cardioprotective effects offered by white grape pomace are evidenced by the prevention of ECG alterations and histological changes in case of isoproterenol-induced myocardial infarction.

This study provides a novel strategy for white grape pomace-enhanced cardio protection, which may be helpful in limiting myocardial damage extension in the case of myocardial infarction. Further studies will have to demonstrate if these observations are valid in human patients.

## Figures and Tables

**Figure 1 ijms-26-02035-f001:**
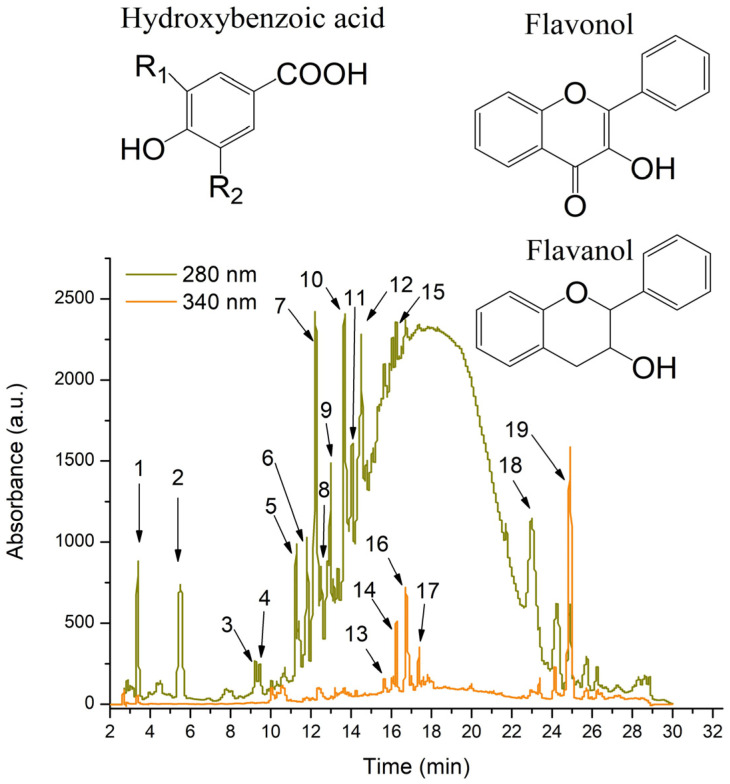
HPLC chromatogram of white grape pomace extract recorded at 280 nm and 340 nm and chemical structures of the standards. The peak identification is detailed in Table 1.

**Figure 2 ijms-26-02035-f002:**
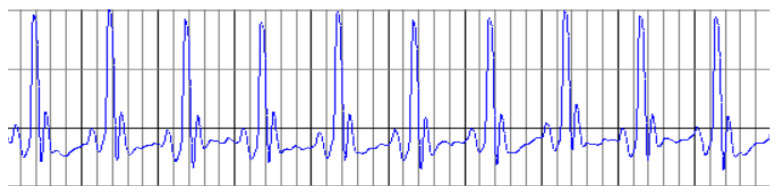
Representative ECG records for all groups were recorded on day 0, at the beginning of the experiment.

**Figure 3 ijms-26-02035-f003:**
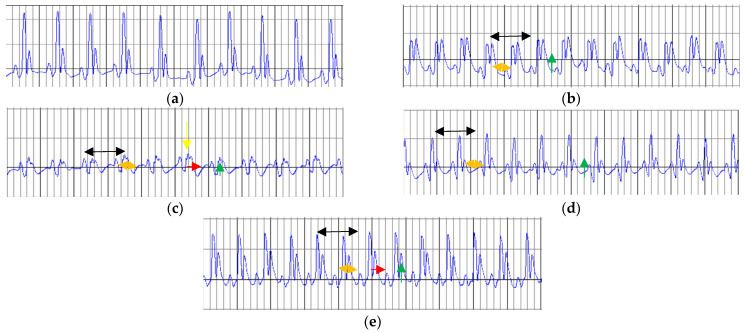
The characteristics of ECG change in each group. (**a**) The negative control group (C) and myocardial infarction groups are presented as follows: (**b**) saline-negative control group (MI_C); (**c**) white grape pomace concentration 1 group (MI_WGP1); (**d**) white grape pomace concentration 2 group (MI_WGP2); and (**e**) Lisinopril-positive control group (MI_LIS). Reduced RR interval (black arrow), prolongation of the QRS complex (orange arrow), increased QT interval (red arrow), ST-segment elevation (green arrow) and reduced amplitude of the R wave (yellow arrow).

**Figure 4 ijms-26-02035-f004:**
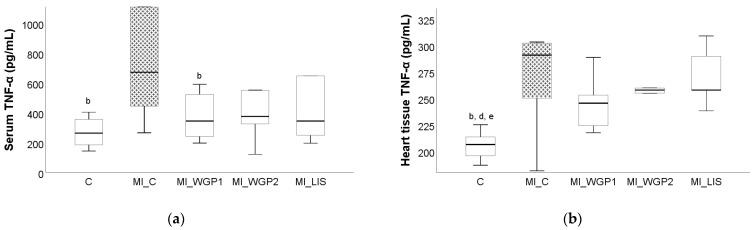
The effect of white grape pomace on tumor necrosis alpha factor (TNF-α) in serum (**a**) and heart tissue (**b**). Rats were grouped into a negative control group (C) and myocardial infarction groups as follows: saline-negative control group (MI_C); white grape pomace concentration 1 group (MI_WGP1); white grape pomace concentration 2 group (MI_WGP2); and Lisinopril-positive control group (MI_LIS), where ^b^ had *p* < 0.05, versus the MI_C group; ^d^ had *p* < 0.05, versus the MI_WGP2; ^e^ had *p* < 0.05, versus the MI_LIS group following Kruskal–Wallis Test.

**Figure 5 ijms-26-02035-f005:**
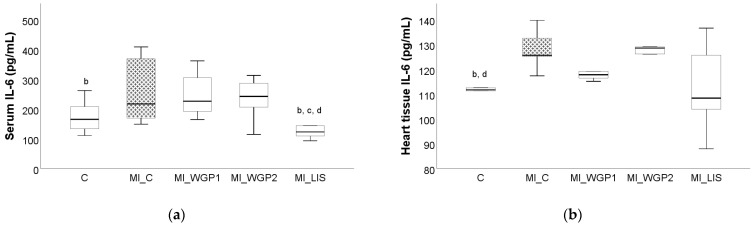
The effect of white grape pomace on interleukin 6 (IL-6) in serum (**a**) and heart tissue (**b**). Rats were grouped into a negative control group (C) and myocardial infarction groups as follows: saline-negative control group (MI_C); white grape pomace concentration 1 group (MI_WGP1); white grape pomace concentration 2 group (MI_WGP2); and Lisinopril-positive control group (MI_LIS). ^b^ *p* < 0.05, versus the MI_C group; ^c^ *p* < 0.05, versus the MI_WGP1; ^d^ *p* < 0.05, versus the MI_WGP2 group following Kruskal–Wallis Test.

**Figure 6 ijms-26-02035-f006:**
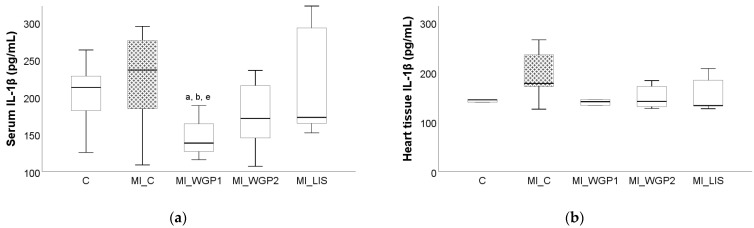
The effect of white grape pomace on interleukin 1β (IL-1β) in serum (**a**) and heart tissue (**b**). Rats were grouped into a negative control group (C) and myocardial infarction groups as follows: saline-negative control group (MI_C); white grape pomace concentration 1 group (MI_WGP1); white grape pomace concentration 2 group (MI_WGP2); and Lisinopril-positive control group (MI_LIS). ^a^ *p* < 0.05, versus the MI_C group; ^b^ *p* < 0.05, versus the MI_WGP1; ^e^ *p* < 0.05, versus the MI_LIS group following Kruskal–Wallis Test.

**Figure 7 ijms-26-02035-f007:**
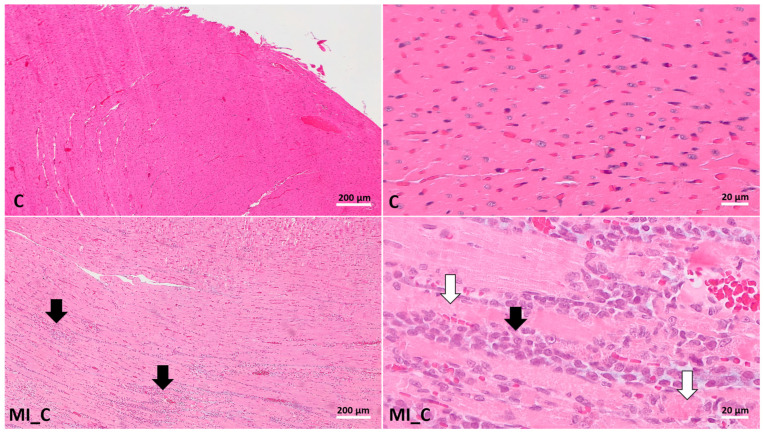
Histological findings in the myocardium of experimental groups. The rats were grouped into the negative control group (C) and positive control group (MI_C), where MI represents myocardial infarction. Compared to normal heart tissue (C), the group MI_C showed multifocal coalescing to extensive myofibre degeneration and necrosis (white arrows) with inflammatory infiltrates (black arrows); hematoxylin and eosin stain.

**Figure 8 ijms-26-02035-f008:**
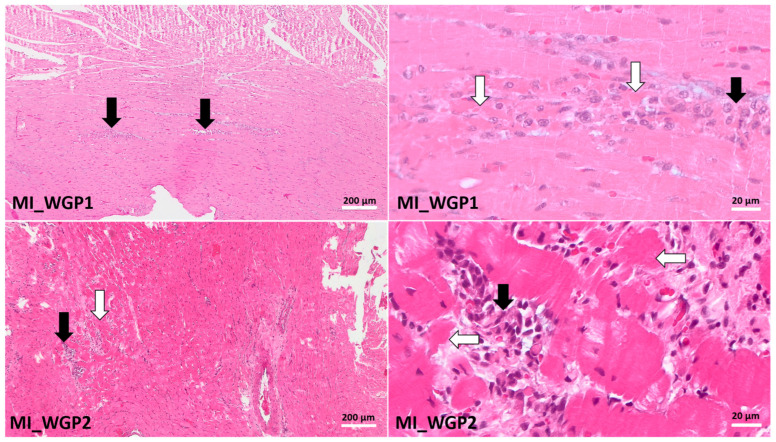
Histological findings in the myocardium of experimentally treated groups. The rats were grouped in MI_WGP1 and MI_WGP2, where MI = myocardial infarction, and WGP = different concentrations of white grape pomace. Group MI_WGP1 showed mild and multifocal extensive myofibre degeneration and necrosis (white arrows) with few inflammatory cells (black arrows). In the group MI_WGP2, the myocardial changes were more prominent than in MI_WGP1, but less severe compared to the positive control group (MI_C); hematoxylin and eosin stain.

**Figure 9 ijms-26-02035-f009:**
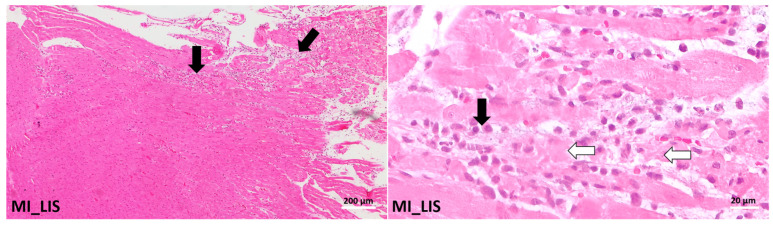
Histological findings in the myocardium of experimentally treated groups. The rats were grouped in MI_LIS, where MI = myocardial infarction, LIS = Lisinopril. Group MI_C showed multi-focal to coalescing to extensive myofibre degeneration and necrosis (white arrows) with inflammatory infiltrates (black arrows); hematoxylin and eosin stain.

**Figure 10 ijms-26-02035-f010:**
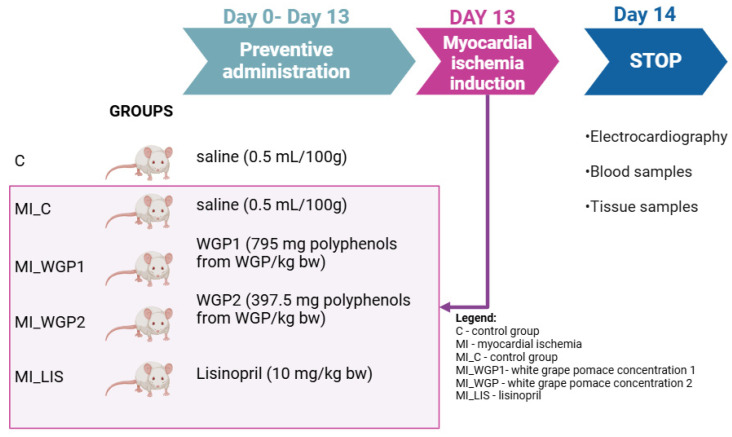
Flowchart demonstrating the study groups and interventions.

**Table 1 ijms-26-02035-t001:** Identification and quantification of the phenolic compounds of the white grape pomace ethanolic extract.

No	R_t_ (min)	UV λ_max_ (nm)	[M+H]^+^ (*m/z*)	Compound	Subclass	Concentration (μg/mL)	References
1	3.29	270	138	Hydroxybenzoic acid	Hydroxybenzoic acid	133.80 ± 0.21	[21]
2	5.96	279	171	Gallic acid	Hydroxybenzoic acid	292.91 ± 18.30	[21]
3	9.23	280	155	Dihydroxybenzoic acid	Hydroxybenzoic acid	68.76 ± 1.98	[22]
4	9.45	260, 290	155	Protocatechuic acid	Hydroxybenzoic acid	75.05 ± 2.16	[22]
5	11.26	280	579, 291	Procyanidin dimer I	Flavanol	416.07 ± 13.04	[21]
6	11.81	280	579, 291	Procyanidin dimer II	Flavanol	408.84 ± 9.68	[21]
7	12.21	280	291	Catechin	Flavanol	1407.33 ± 22.82	[21]
8	12.51	280	1155, 291	Procyanidin tetramer	Flavanol	224.27 ± 3.94	[23]
9	12.97	280	579, 291	Procyanidin dimer III	Flavanol	406.15 ± 6.73	[23]
10	13.66	280	291	Epicatechin	Flavanol	1083.30 ± 21.46	[21]
11	14.05	2803	867, 291	Procyanidin trimer	Flavanol	268.54 ± 7.63	[23]
12	14.50	280	579, 291	Procyanidin dimer IV	Flavanol	1026.22 ± 15.65	[23]
13	15.67	261, 355	611, 303	Quercetin-rutinoside	Flavonols	55.87 ± 2.02	[21]
14	16.23	263, 355	465, 303	Quercetin-glucoside	Flavonols	126.30 ± 2.73	[21]
15	16.30	280	443	Epicatechin gallate	Flavanol	70.16 ± 1.16	[24]
16	16.72	262, 355	479, 303	Quercetin-glucuronide	Flavonols	295.59 ± 9.35	[23]
17	17.33	253, 350	449	Kaempferol-glucoside	Flavonols	117.68 ± 1.93	[21]
18	22.91	280	422, 346	NI	Tannin	825.69 ± 13.96	[25]
19	24.88	370	764, 548	NI	Tannin	574.12 ± 3.68	[25]
	Total					7876.71 ± 158.41	

Note: Values are expressed as mean ± SD (*n* = 3).

**Table 2 ijms-26-02035-t002:** ECG parameters were measured on day 0, at the beginning of the experiment for all groups.

Groups	RR (mS)	HR (b/min)	PR (ms)	QRS (ms)	QT (ms)	QTc (ms)	AmpR
C	202.0 (194.0–210.0)	297.0 (285.7–309.2)	43.5 (42.0–47.0)	39.0 (37.0–40.0)	87.0 (71.0–88.0)	74.5 (60.2–79.4)	2.5 (2.0–3.0)
MI_C	196.5 (188.0–213.0)	305.5 (281.6–319.1)	46.0 (43.0–51.0)	38.0 (36.0–43.0)	81.0 (76.0–87.0)	70.1 (66.2–73.3)	2.0 (2.0–3.0)
MI_WGP1	201.0 (190.0–212.0)	298.5 (283.0 –315.7)	48.5 (43.0–51.0)	38.0 (36.0–42.0)	80.0 (74.0–85.0)	70.3 (66.1–71.1)	3.0 (2.0–3.0)
MI_WGP2	198.5 (187.0–224.0)	302.3 (267.8–320.8)	48.5 (44.0–50.0)	41.0 (39.0–44.0)	81.5 (76.0–84.0)	67.8 (66.2–72.5)	2.5 (2.0–3.0)
MI_LIS	199.0 (187.0–224.0)	301.5 (267.8–320.8)	43.5 (41.0–46.0)	39.0 (36.0–41.0)	79.5 (74.0–87.0)	69.6 (63.9–72.7)	2.0 (2.0–3.0)

Rats were grouped into a negative control group (C) and myocardial infarction groups as follows: saline-negative control group (MI_C); white grape pomace concentration 1 group (MI_WGP1); white grape pomace concentration 2 group (MI_WGP2); and Lisinopril-positive control group (MI_LIS). Values are presented as median (25–75 percentiles). According to the Kruskal–Wallis Test, there was no statistical significance between groups on day 0.

**Table 3 ijms-26-02035-t003:** ECG parameters were measured one day after myocardial infarction induction for all groups.

Groups	RR (mS)	HR (b/min)	PR (ms)	QRS (ms)	QT (ms)	QTc (ms)	AmpR
C	196.5 (186.0–224.0)	305.4 ^b,c,d,e^ (267.8–322.5)	46.5 (44.0–49.0)	39.0 ^b,c,d,e^ (36.0–40.0)	80.0 ^b,e^ (74.0–87.0)	70.5 ^b,c,d,e^ (66.3–71.3)	3.0 ^b,c,d,e^ (2.0–3.0)
MI_C	157.0 ^a^ (151.0–165.0)	382.2 (363.6–397.3)	43.5 (40.0–47.0)	46.05 (44.0–53.0)	93.0 (89.0–101.0)	91.5 (87.8–97.4)	0.0 (0.0–0.0)
MI_WGP1	159.0 ^a^ (142.0–168.0)	377.4 (357.1–422.5)	42.5 (39.0–47.0)	50.5 (46.0–52.0)	86.5 (82.0–89.0)	84.6 (80.7–87.8)	1.0 ^b^ (1.0–1.0)
MI_WGP2	158.5 ^a^ (142.0–162.0)	378.5 (370.3–422.5)	44.0 (43.0–45.0)	46.5 (44.0–55.0)	81.0 ^e^ (78.0–85.0)	78.0 ^b^ (76.0–78.9)	1.0 ^b^ (1.0–2.0)
MI_LIS	154.0 ^a^ (148.0–172.0)	389.6 (348.8–405.4)	42.5 (40.0–46.0)	51.5 (43.0–54.0)	90.0 (83.0–95.0)	87.9 (80.7–91.0)	1.0 ^b^ (1.0–1.0)

Rats were grouped into a negative control group (C) and myocardial infarction groups as follows: saline-negative control group (MI_C); white grape pomace concentration 1 group (MI_WGP1); white grape pomace concentration 2 group (MI_WGP2); and Lisinopril-positive control group (MI_LIS). Values are presented as median (25–75 percentiles) where ^a^ had *p* < 0.05, versus the C group; ^b^ had *p* < 0.05, versus the MI_C group; ^c^ had *p* < 0.05, versus the MI_WGP1; ^d^ had *p* < 0.05, versus the MI_WGP2; ^e^ had *p* < 0.05, versus the MI_LIS group following Kruskal–Wallis Test.

**Table 4 ijms-26-02035-t004:** In vivo antioxidant activity of the white grape pomace.

Groups	TAC (mmol Trolox eq/L)	TOS (mmol H_2_O_2_/eq/L)	NO (mmol/L)	MDA (nmol/L)	Total Thiols (mmol/L)	OSI
C	1096 (1.093–1.100)	7.7 ^b,c,e^ (6.7–8.5)	39.5 ^b^ (38.7–43.6)	3.4 ^b,c,e^ (3.2–3.6)	368.0 (267.0–587.0)	7.0 ^b,c,e^ (6.1–7.7)
MI_C	1097 (1.095–1.099)	10.7 (9.7–12.2)	46.2 (44.5–50.8)	4.2 (3.3–4.5)	319.0 ^c,e^ (297.0–395.0)	9.7 (8.8–12.1)
MI_WGP1	1095 (1.094–1.096)	13.0 (11.8–13.7)	36.2 ^b^ (34.2–40.0)	4.0 ^a^ (3.8–4.3)	438.0 (405.0–505.0)	11.8 (10.7–12.5)
MI_WGP2	1090 ^a,b,c,e^ (1.089–1.091)	5.9 ^b,c,e^ (5.3–6.4)	30.0 ^b,e^ (28.0–38.0)	3.6 ^c,e^ (3.5–3.9)	275.0 ^a,c,e^ (250.0–309.0)	5.4 ^b,c,e^ (4.9–5.9)
MI_LIS	1093 ^b^ (1.092–1.095)	15.2 (8.0–17.6)	40.3 ^b^ (39.0–43.1)	4.3 (4.1–4.7)	415.0 (397.0–481.0)	13.8 (7.3–16.1)

Rats were grouped into a negative control group (C) and myocardial infarction groups as follows: saline-negative control group (MI_C); white grape pomace concentration 1 group (MI_WGP1); white grape pomace concentration 2 group (MI_WGP2); and Lisinopril-positive control group (MI_LIS). Values are presented as median (25–75 percentiles) where ^a^ had *p* < 0.05, versus the C group; ^b^ had *p* < 0.05, versus the MI_C group; ^c^ had *p* < 0.05, versus the MI_WGP1; ^e^ had *p* < 0.05, versus the MI_LIS group following Kruskal–Wallis Test.

## Data Availability

Data are available on request.

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
