# Peer review of "Anti-Inflammatory and Antioxidant Effects of White Grape Pomace Polyphenols on Isoproterenol-Induced Myocardial Infarction"

_ijms, 2025, doi:10.3390/ijms26052035_

Round 1

Reviewer 1 Report

Comments and Suggestions for Authors

The research article concerns the assessment of the effect of white grape pomace on electrocardiogram parameters, oxidative stress, and pro-inflammatory cytokines in isoproterenol (ISO)-induced myocardial infarction in rats. The main topic of the article is highly interesing and align well with the aims and scope of the journal.

However, several issues need to be addressed before the manuscript can be considered for publication.

Major issues:

  • The quantification of phenolic compounds is not described. How was this performed? Please provide more information, including the calibration curve used for quantification. The reference materials used for compound identification are not included in Section 4.2. Moreover, it would be useful to cite the references in Table 1 or below it, as you mention in line 472 that reference data were used for identification. Furthremore, in Table 1, concentration values should be expressed as mean ± SD; currently the standard deviation is missing;
  • In section 4.5, you report that total polyphenol content was analyzed in triplicate, with results expressed as mean ± SD. However, these results do not appear to be included in the text. Please clarify and ensure they are properly reported.

    Minor Issues:
  • The reference formatting does not comply with MDPI guidelines. Please correct this accordingly.
  • Avoid the use of "we" or "our" (e.g., in linea 93 and 304) and adopt a more impersonal tone throughout the manuscript.
  • Table 1: "Tanin" should be corrected to "Tannin"
  • Section 4.2: Please include all reagents used in this study, such as, for example, ethanol for grape pomace extraction as well as the reference materials used.

    Overall, the study presents relevant and valuable findings, but the issues outlined above should be addressed to enhance the clarity and scientific rigor of the manuscript.
Comments on the Quality of English Language

The English could be improved, there are some typos and some parts would benefit from a more impersonal tone.

Author Response

Response Letter

Before addressing each comment below, the authors thank the reviewers for their time and valuable comments.

Response to Reviewer’s comments

Reviewer 1

The research article concerns the assessment of the effect of white grape pomace on electrocardiogram parameters, oxidative stress, and pro-inflammatory cytokines in isoproterenol (ISO)-induced myocardial infarction in rats. The main topic of the article is highly interesing and align well with the aims and scope of the journal.

R: Thank you very much for your appreciation!

However, several issues need to be addressed before the manuscript can be considered for publication.

Major issues:

The quantification of phenolic compounds is not described. How was this performed? Please provide more information, including the calibration curve used for quantification.

R: Thank you very much for your suggestions. We have added the details in the materials and methods section. We included the calibration curve details.

The reference materials used for compound identification are not included in Section 4.2.

R: We thank you for your observation. We have added the missing details.

Moreover, it would be useful to cite the references in Table 1 or below it, as you mention in line 472 that reference data were used for identification.

R: We thank you for your suggestion. We have added the references in a new column integrated into the table.

 Furthremore, in Table 1, concentration values should be expressed as mean ± SD; currently, the standard deviation is missing;

R: We thank you for your observation. We apologize for this error. We have added the missing ± SD.

In section 4.5, you report that total polyphenol content was analyzed in triplicate, with results expressed as mean ± SD. However, these results do not appear to be included in the text. Please clarify and ensure they are properly reported.

R: Thank you very much for your observation. The results were inserted in the text.

Minor Issues:

The reference formatting does not comply with MDPI guidelines. Please correct this accordingly.

R: Thank you very much for your observation. All refences were formatted according to MDPI guidelines.

Avoid the use of "we" or "our" (e.g., in linea 93 and 304) and adopt a more impersonal tone throughout the manuscript.

R: Thank you very much for your observation. The whole manuscript was revised and an impersonal ton was used, as suggested.

Table 1: "Tanin" should be corrected to "Tannin"

R: Thank you very much for your observation. The error was corrected.

Section 4.2: Please include all reagents used in this study, such as, for example, ethanol for grape pomace extraction as well as the reference materials used.

R: Thank you very much for your recommendation. We have added the reagents in the section 4.2.

Overall, the study presents relevant and valuable findings, but the issues outlined above should be addressed to enhance the clarity and scientific rigor of the manuscript.

R: Thank you again for helping us to improve our manuscript.

Comments on the Quality of English Language

The English could be improved, there are some typos and some parts would benefit from a more impersonal tone.

R: Thank you very much for your recommendation! The manuscript was revised by a proficient English speaker.

Reviewer 2 Report

Comments and Suggestions for Authors

Some comments should be addressed as follows:

  1. Abstract: Line 39: it should be modified to clearly describe the aim of this study. Line 42: please avoid abbreviations in the first of sentence. The full expression of abbreviations should be spelled out; for instance, MDA, NO, and so on. The authors should briefly outline the approaches conducted in this study. Key findings should be highlighted.
  2. Intro: the aim of this study should be delineated in detail at the end of this section.
  3. Results: the authors are encouraged to incorporate the HPLC chromatogram in the manuscript and draw the chemical structures of compounds. Line 124-127: please rewrite since it is along sentence. Line 128: please replace (did not influence) with (showed no impact ………..). Line 130: please rectify it. Please add significant differences in table 2, 3, and 4. For the oxidative stress parameters: the authors should measure other oxidative stress parameters; for instance, SOD, catalase, GSH, GST, and GPx, in addition to Nrf2 since the oxidative parameters are not enough to emphasize the antioxidant activity of white grape pomace. For the inflammatory biomarkers, the authors should measure NF-κB. In Fig. 3, 4, and 5, please add multiple comparison and significant differences between experimental groups. Fig. 6: do you have magnified section to clearly illustrate extensive myofiber degeneration and necrosis with inflammatory infiltrates? Scale bars are invisible in Fig. 6-8. Please check these references to improve the quality of the manuscript (https://doi.org/10.3390/pharmaceutics15092306; https://doi.org/10.3390/catal13020443).  
  4. Discussion: please describe the aim of this study. Line 300-302: the authors should evaluate these factors to emphasize this argument. The antioxidant and anti-inflammatory mechanism of WGP extract should be discussed in light of data obtained in this study.
  5. Limitations: the authors should mention the lack of protein analysis in terms of WB, IHC and molecular mechanism, which could be addressed in the future studies.
  6. Methods: How many rats were assigned to each group? The catalogue number of each kit should be added, company, and country. Tissue homogenates preparation: it should be dragged to after collection of blood samples and before the evaluation of oxidative stress and inflammatory biomarkers. Did you apply ANOVA for statistical analysis? Please add a scheme for in vivo experimental design.
Comments on the Quality of English Language

The authors should enhance the quality of writing.

Author Response

Response Letter

Before addressing each comment below, the authors thank the reviewers for their time and valuable comments.

Response to Reviewer’s comments

Some comments should be addressed as follows:

Abstract: Line 39: it should be modified to clearly describe the aim of this study.

R: Thank you for your suggestions. We modified the abstract so that the aim was better described.

Line 42: please avoid abbreviations in the first of sentence. The full expression of abbreviations should be spelled out; for instance, MDA, NO, and so on.

R: Thank you for your observation. We also defined all abbreviations at their first appearance in text and we avoided using abbreviations at the beginning of the sentence.

The authors should briefly outline the approaches conducted in this study. Key findings should be highlighted.

R: Thank you for your suggestions. The methods were better explained and key findings were highlighted as suggested.  

Intro: the aim of this study should be delineated in detail at the end of this section.

R: Thank you for your observation. We rephrased the last paragraph from the introduction to better highlight the aim of this study.

Results: the authors are encouraged to incorporate the HPLC chromatogram in the manuscript and draw the chemical structures of compounds.

R: Thank you for your observation. We incorporated the HPLC chromatograms and added the chemical structures of the standards by which the compounds were quantified to avoid an overloaded graph.

Line 124-127: please rewrite since it is along sentence. Line 128: please replace (did not influence) with (showed no impact ………..). Line 130: please rectify it.

R: Thank you for your suggestions. The phrase from lines 124-127  and 130 were rephrased and replacement was done in line 128.

Please add significant differences in table 2, 3, and 4.

R: Thank you for your observation. For Table 2, according to the Kruskal-Wallis Test, there was no statistical significance between groups on day 0. This detail was added to the text, below the table. The significance between the groups was indicated through the letters a, b, c, d, and e. It was inserted below each row so it was indeed difficult to be observed. We have now modified and inserted more comprehensively. We thank you for your observation that helped improve data reporting.

For the oxidative stress parameters: the authors should measure other oxidative stress parameters; for instance, SOD, catalase, GSH, GST, and GPx, in addition to Nrf2 since the oxidative parameters are not enough to emphasize the antioxidant activity of white grape pomace.

R: Thank you for your observation. We agree that evaluation of additional oxidative stress parameters such as SOD, catalase, GSH, GST, and GPx, and activity of Nrf2 and evaluation of NF-κB as an inflammatory biomarker would bring incontestable value to our manuscript, but at the moment it is impossible for us (in the next 10 days) to obtain additional kits to perform the suggested measurements. We will take into consideration your valuable suggestions to be used in our future research and mention it in “Limitations and Future Directions”.

For the inflammatory biomarkers, the authors should measure NF-κB.

R: We thank you for your suggestion. We will take into consideration your valuable suggestions to be used in our future research and mention them in “Limitations and Future Directions”.

In Fig. 3, 4, and 5, please add multiple comparisons and significant differences between experimental groups.

R: We thank you for your observation. We have added the statistical differences between the experimental groups in the text as well and highlighted them in red.

Fig. 6: do you have magnified section to clearly illustrate extensive myofiber degeneration and necrosis with inflammatory infiltrates? Scale bars are invisible in Fig. 6-8. Please check these references to improve the quality of the manuscript (https://doi.org/10.3390/pharmaceutics15092306; https://doi.org/10.3390/catal13020443). 

R: Thank you for the observation. We have made the improvements as suggested.

Discussion: Please describe the aim of this study. Line 300-302: the authors should evaluate these factors to emphasize this argument. The antioxidant and anti-inflammatory mechanism of WGP extract should be discussed in light of data obtained in this study.

R: Thank you for your observations. The aim of the study was mentioned in the introduction. At the beginning of the discussion, we highlighted the main findings according to the aim of the study. Please see:

“The present study showed that WGP pre-treatment offered cardioprotective effects in isoproterenol-induced MI. Specifically, it helped in controlling the evaluated oxidative stress parameters and pro-inflammatory cytokines. It also helped in reducing the electrocardiographic and histological changes associated with MI.”

Also: “It is well known that flavanols interact with cell membranes by modulating fluidity and permeability, primarily through their hydroxyl-substituted aromatic rings (35). Their protective effects against oxidative stress involve inhibiting lipid peroxidation, scavenging free radicals, and chelating redox-active metals, thereby preserving membrane integrity and function (36–38). Also, these compounds exert anti-inflammatory effects by modulating key signaling pathways involved in the inflammatory response, such as nuclear factor kappa-light-chain-enhancer of activated B cells (NF-κB) and mitogen-activated protein kinase (MAPK), leading to the downregulation of pro-inflammatory cytokines and mediators (39).”

“In the present study, WGP protected against oxidative stress by stimulating thiol production when administered in higher concentrations….

“Similar to the results from the present study, Balea et al. reported that red wine grape pomace reduced serum levels of MDA in ISO-induced myocardial ischemia (14)….

 “The administration of grape pomace extracts has been previously associated with reduced levels of malondialdehyde, indicating a decrease in oxidative stress (66), and in this paper, this effect was observed in MI….

“In the present study, WGP administration was observed to reduce TNF-α and IL-1β levels and had limited effects on IL-6. The underlying mechanisms by which grape pomace influences TNF-α levels involve the modulation of key inflammatory pathways….

“…white and red pomace extracts it was observed that even though a decreasing trend in the level of IL-6 was obtained for both types, no statistical difference was reached (79). These results are consistent with the findings of this study….

Limitations: the authors should mention the lack of protein analysis in terms of WB, IHC and molecular mechanism, which could be addressed in the future studies.

R: Thank you for your suggestions! The limitation was presented in the section: ”Additionally, molecular evaluations of proteins involved in pathways related to myocardial inflammation, fibrosis, and necrosis were not performed, though they would have provided valuable insights”

Methods: How many rats were assigned to each group?

R: Thank you for your observations and suggestions. It was already mentioned in the manuscript that “The rats were randomly divided into five groups of ten animals each”.

The catalogue number of each kit should be added, company, and country.

We have added the missing details.

Tissue homogenates preparation: it should be dragged to after collection of blood samples and before the evaluation of oxidative stress and inflammatory biomarkers.

We reorganized the Material and Methods section according to your suggestions.

Did you apply ANOVA for statistical analysis? Please add a scheme for in vivo experimental design.

We used Kruskal-Wallis Test. This test extends the Mann-Whitney U test to more than 2 groups, and it is the non-parametric equivalent of the Analysis of Variance (ANOVA).

Comments on the Quality of English Language

The authors should enhance the quality of writing.

R: Thank you very much for your recommandation! The manuscript was revied by a profiecency English speaker.

Reviewer 3 Report

Comments and Suggestions for Authors

In this manuscript, the authors examined the effects of white grape pomace (WGP) extract on electrocardiograms, oxidative stress, and pro-inflammatory cytokines in a rat model of myocardial infarction (MI) induced by isoproterenol (ISO). The results indicate that administration of WGP for 14 days provided a cardioprotective effect and mitigated the prolongation of both QT and QTc intervals. Additionally, WGP was associated with reduced serum levels of nitric oxide (NO) and malondialdehyde (MDA), and it led to a decrease in tumor necrosis factor-alpha (TNF-α), interleukin-6 (IL-6), and interleukin-1 beta (IL-1β).While the selection of topics for this study is commendable and the results are promising, several issues remain that require attention, and certain aspects could be enhanced. Consequently, I recommend major revisions before accepting this manuscript.

  1. It is advisable to provide the specific doses for MI_WGP1 and MI_WGP2 in parentheses upon their initial mention, facilitating comparisons for readers.

  1. An explanation is needed for the observed discrepancy in dosage between MI_WGP1 and MI_WGP2, where the latter is twice the former; however, the activity levels between the two groups did not demonstrate significant differences and lacked a dose-dependent relationship in multiple tests.

  1. Given that WGP contains a variety of polyphenolic compounds, it is recommended to conduct preliminary experiments to identify which specific ingredient or ingredients may be contributing most significantly to its medicinal effects.

Author Response

Response Letter

Before addressing each comment below, the authors thank the reviewers for their time and valuable comments.

Response to Reviewer’s comments

Reviewer 3

In this manuscript, the authors examined the effects of white grape pomace (WGP) extract on electrocardiograms, oxidative stress, and pro-inflammatory cytokines in a rat model of myocardial infarction (MI) induced by isoproterenol (ISO). The results indicate that administration of WGP for 14 days provided a cardioprotective effect and mitigated the prolongation of both QT and QTc intervals. Additionally, WGP was associated with reduced serum levels of nitric oxide (NO) and malondialdehyde (MDA), and it led to a decrease in tumor necrosis factor-alpha (TNF-α), interleukin-6 (IL-6), and interleukin-1 beta (IL-1β).While the selection of topics for this study is commendable and the results are promising, several issues remain that require attention, and certain aspects could be enhanced. Consequently, I recommend major revisions before accepting this manuscript.

It is advisable to provide the specific doses for MI_WGP1 and MI_WGP2 in parentheses upon their initial mention, facilitating comparisons for readers.

 R: Thank you very much for your observation! Specific doses were added upon their initial mention in the manuscript.

An explanation is needed for the observed discrepancy in dosage between MI_WGP1 and MI_WGP2, where the latter is twice the former; however, the activity levels between the two groups did not demonstrate significant differences and lacked a dose-dependent relationship in multiple tests.

 R: Thank you very much for your remark. MI_WGP2 received half of the dose from MI_WGP1. We have now added this information and also in the schematic figure that was newly introduced in the materials and methods section. 

 Regarding the dose-response relationship, in the in vivo animal studies with plant extracts, a clear dose-dependent relationship is not always observed. One key explanation is that the selected doses may fall within a range on the dose-response curve where the slope is minimal—meaning that incremental changes in dose do not lead to substantial changes in effect. However, several additional factors can contribute to this phenomenon. One of them is that plant extracts are complex mixtures of multiple bioactive compounds. These compounds can interact synergistically or antagonistically, making it challenging to predict or observe a straightforward, linear dose-response relationship. Next, pharmacokinetic variability may also affect the in vivo responses. The processes of absorption, distribution, metabolism, and excretion (ADME) can vary significantly. These factors can alter the actual concentration of active compounds reaching the target site, thus blunting or distorting a dose-dependent effect. Another one will take into account that many biological systems exhibit receptor or enzyme saturation. At a certain point, increasing the dose further may not increase the effect because the target sites (receptors, enzymes, etc.) are fully occupied or the metabolic pathways are maxed out. Also, there might be a narrow therapeutic window, like in the case of some drugs. For some extracts, there may be a narrow range of effective doses. Outside this range, the effect may be minimal or counterproductive, making it appear as if there is no dose dependency when in fact the optimal window was missed.

Overall, while the chosen concentration range is a critical factor, these additional biological and experimental complexities often contribute to the lack of a straightforward dose-response relationship in vivo studies of plant extracts.

Given that WGP contains a variety of polyphenolic compounds, it is recommended to conduct preliminary experiments to identify which specific ingredient or ingredients may be contributing most significantly to its medicinal effects.

R: Thank you very much for your suggestion! We totally agree that further research is necessary to identify which specific ingredient or ingredients may be contributing most significantly to its medicinal effects

Round 2

Reviewer 2 Report

Comments and Suggestions for Authors

I am satisfied with the authors' response. They have carefully addressed all previous claims, and their explanation and changes to the manuscript make the current study clearer for readers.

Reviewer 3 Report

Comments and Suggestions for Authors

Most of the issues were addressed.